# Mapping of Greek Marine Finfish Farms and Their Potential Impact on the Marine Environment

George Katselis [ID], Konstantinos Tsolakos *[ID] and John A. Theodorou [ID]

Department of Animal Production, Fisheries and Aquaculture, School of Agricultural Sciences, Patras University, 30200 Mesolongi, Greece; gkatselis@upatras.gr (G.K.); jtheo@upatras.gr (J.A.T.)
* Correspondence: k.tsolakos@upatras.gr

**Abstract:** The Greek marine aquaculture has a leading role in the Mediterranean mariculture industry, mainly in the export sector and its prominent contribution to the country's economy. In the present study, the spatial distribution of Greek finfish farming activity and its potential impact zones on the marine environment were estimated and mapped. The Greek coastline was scanned via Google Earth satellite images for the period of June 2016 to May 2017, with 433 fish farm cage arrays being detected. For each cage array, the zones at different distances corresponding to various type of impacts were mapped by means of GIS technologies. Seventy-five spatial clusters of cage arrays (sc) were revealed, including cage arrays with shown connectivity. As per the findings, Greek marine fish farming activity shows a high level of spatial aggregation but with a relative moderate intensity of impacts due to legal constraints, which play a crucial role in controlling the spatial distribution of activity at a local, regional, and national scale. The results reflect an important source of geodata, necessary for the spatial planning of activity, the monitoring of environmental impacts, and the research itself.

**Keywords:** fish farm; satellite image; aquaculture waste; environmental impacts

## 1. Introduction

The increased demand for fishery products with high nutritional value, along with the decline in fishery production currently available at higher costs, has led to the rapid global growth of fish farming [1–4]. However, as aquaculture activity has increased worldwide, competition for space and water quality problems at a local level have resulted, which has also resulted in a negative public perception of mariculture's environmental and aesthetic impacts [5].

The impact of marine fish farms refers to the total change in the bentho-community and nutrient enhancement of the water column under and near cages, with reduced footprints as the distance increases, which is traceable at distances of 500–1000 m [6–12]. Similarly, at a distance not far from farms, aggregations of wild fishes have been recorded, with the effect of farms on wild fish's demography being traceable at distances of 3–6 km [13–18] (Table 1).

**Table 1.** The environmental impacts of fish farming in relation to the distance from the cage's edges.

| Distance from the Cages (m) | Sediment | Water Column | Biodiversity | Wild Stocks |
|---|---|---|---|---|
| 0 (under the cage) | Anoxic sediment beneath the cages (methane CH4 and hydrogen sulphide H2S) [9]. Enrichment in organic matter (8%) at the level of the deepest layer of sediment (10–15 cm) [19]. | High values of the organic matter ratio (>50%) and organic carbon and nitrogen contents (>31 and >4.7 mg/g, respectively) were found within 24 m of the cages [20]. | Unacceptable status of all ecological indicators (BQI-species; BQI-family; H′; BENTIX; M-AMBI; BOPA; BENTIXfamily) [7]. | Sparids, Carangids, and Mugilids show the greatest fish diversity beneath sea cages [21]. *Boops boops* and *Pollachius virens* are the most abundant fish species beneath sea cages [21,22]. |
| 25 | TOC and TON values, oxygen consumption, and PO4 release are higher than the control station [11]. The concentrations of most elements (organic matter; Cu, Zn, P, U and coarse sediment; Pd, Pb, Sr, Mg, Ca, Na) directly beneath and close to the fish cages were significantly higher [12]. | Organic matter ratio and organic carbon and nitrogen contents are reduced to ca. 16%, 4.0 mg/g and 0.6 mg/g at a site 200 m from the fish cage while high acid volatile sulfide values indicated enrichment effects [20]. | BENTIX and M-AMBI indicators (28% and 44%, respectively) vs. 83% control station [7]. Significant effect on macrofauna [9]. | Greatest concentrations of wild fish occurred immediately beneath farms, with a steep decline in the abundance of fish just 10–100s of meters away [22]. |
| 50 | | | BENTIX and M-AMBI indicators (33%, 56%) vs. 83% at the control station [7]. Increased macrofaunal abundance [6]. | |
| 100 | Sedimentation rates 5–10% compared with the rates just beneath the cages [23]. | | | |
| 200 | Contaminant levels for Zn and Cu approaching background levels at distances greater than 200 m from the original cage locations [10]. Spatial extent of waste dispersal [20]. | | Increase in the meiofaunal abundance [23]. Significant variation in the abundance of polychaetes and amphipod families decreased significantly in number and diversity [24]. | Abundance (52 to 2837×), biomass (2.8 to 1126×), and number of species (1.6 to 14×) were greater in fish farms than control sites 200 m from farms [25]. |
| 500 1000 3000 | Sedimentation rate 40% higher than the control station [23]. | Dispersal of waste. Particulate organic matter (POM), mean isotopic values of carbon (δ13C), and nitrogen (δ15N); significant differences between distance categories (0; 500; 1000) [26]. | | Fish farms are connected through attraction to wild fish populations (transmission of diseases and parasites, change in genetic cohesion (hybridization) and reduce the survival capacity of wild stocks [13,15–18,27,28]. |

Marine spatial planning is about managing the distribution of human activities in space and time to achieve ecological, economic, and social objectives and outcomes [29]. A Geographical Information System (GIS) is an automated information system that is able to compile, store, retrieve, analyze, and display mapped spatial data, with applications to a wide spectrum of activities [30], including aquaculture [31–38] and fishery, providing significant possibilities for its management [39] with increased use during the last decades by government officials, natural resource analysts, and many others. During the last decade, based on GIS and remote sensing methodologies, aquaculture has been the subject of numerous studies to identify suitable sites for its development, resolve complex environmental and socioeconomic constraints, monitor water quality, assess fish farming's environmental impact, etc. [38].

Marine aquaculture in Greek coastal waters was introduced in the early 1980s [40], with Turkey nowadays being one of the leaders in the Mediterranean mariculture industry specializing in the production of euryhaline finfish species, such as the Mediterranean Sea bass (*Dicentrarchus labrax*) and gilthead sea bream (*Sparus aurata*). Annual production during the period of 2010–2019 ranged from 85 to 105 kt, with an estimated value at 0.55 to 0.65 billion $ [41]. It is an export-oriented product in major EU markets [42] while numerous fish farms participate in the ASC's (Aquaculture Stewardship Council's) aquaculture certification program to achieve recognition and gain a responsible aquaculture reward [43], thus increasing their competitive advantage. On the other hand, specific environmental quality standards, as defined by Greek legislation in the zone up to 100 m around cage edges, are allowed in combination with a monitoring program (121634/7242/20 December 2019 newsletter of Ministry Environmental and Energy).

Despite the country's economic footprint and the requirement for activity application, the authorities and managers currently do not have a GIS database about the activity. The available georeferenced information is limited and refers to the location of farms [31,37,44,45] and the estimation of environmental impacts at zones up to 3000 m [45] or at zones up to fish farm waste dispersion (≈100 m) around the cage arrays according to climate change [37]. The framework legislation (Common Ministerial Decision No 31722/2011, FEK 2505 ratified on 4 November 2011) provides guidelines for industry development, including described spatial planning and the participation of stakeholders to make decisions at a local and regional level during the licensing stage [37]. A friction point in the licensing stage debate is fish farms' impacts, where the spatial information (or simulations) about impacts is fuzzy, exacerbating the controversy and decelerating the licensing completion.

The Growth Vision 2030 for Greek marine fish aquaculture, as formulated since 2012, is to double its production by 2030 [42], demanding, however, more space, and an increase in the spatial information and development of modern planning tools. The present study provides a detailed analysis of marine fish farms on the Greek coastline and the spatial distribution of the possible environmental impacts and mapping. The produced vectors improve the available spatial information about fish farms' impacts toward sustainable marine spatial planning, local development projects, and conservation studies.

## 2. Materials and Methods

The Greek coastline was scanned via Google Earth satellite images for the period of June 2016 to May 2017 with fish farm cage arrays being detected. Cages with a distance between them lower than 20 m consist a cage array (thereafter *l*).

In each *l* recorded on the satellite images in the period 2016–2017, the number, type (circular: *NC* and square: *NS*), and dimensions (diameter and side for circular and square cages, respectively) were recorded. The arcsine-transformation of the ratio of *NC* to the total cages of *l* ($AsNC_l$) was estimated [46]:

$$AsNC_l = Arcsine\left(\sqrt{\frac{NC_l}{NC_l + NS_l}}\right) \qquad (1)$$

For each *l*, the cages' upper surface (*SA*) and the functional farming volume (*V*) (thereafter farming volume) was estimated as:

$$SA_l = \sum_{k=0}^{x} a_{l,k}^2 + \sum_{k1=0}^{x1} \pi \left(\frac{d_{l,k1}}{2}\right)^2 \text{ and } V_l = SA_l * dp \qquad (2)$$

where *k* is the square cages; *k*1 is the circular cages; *x* and *x*1 the number of square and circular cages, respectively; *a* is the side of the square and *d* is the diameter of the circular cage, respectively; and *dp* is the cage's functional depth for farming, which is defined as 10 m.

A preliminary analysis of $V_l$ of 123 cage arrays, where the available satellite images cover more than 2 times during the period of June 2016 to May 2017, showed that the CV% (100 standard deviation/mean) ranged from 0–37.5% with a mean value of 3.27%, indicating very low seasonal variation of $V_l$.

With a polygon around the *l* outside of the edges that included the cages, the *l* was georeferenced while the included area on the polygon was defined as the area of *l* ($Area_l$). The georeferenced data was mapped by GIS software (QGIS ver. 3.16.2) [47].

The buffer area ($cArea_i^*$) created by one or more cage arrays for each category *i*, ignoring the individually overlapping areas that originated from each cage array (i: 0, 25, 100, 200, 500, 1000, and 3000 m are the distances from *l*'s edge) ($cArea_i^*$: common area of *l*'s for distance i), was designed. From the layer $cArea_i^*$, the removed area that corresponded to land provided the $cArea_i$ that corresponded to the sea area of each impact i. For each *l*, the functional volume per $cArea_i$ ($Vs_{l,i}$) where the *l* participated in its creation was calculated. *Vs* is the farming volume per unit of the impacted area while it expresses the level of farming pressure on the impacted area, which can be used as a measure of the impact's intensity. The shorter distance of the cage array from the shoreline (Dst) was estimated as the shorter distance of the $cArea_0$ layer from the land layer. The land layer was provided by the European Environment Agency [48] (accuracy 50 m).

For each *l*, a set of variables (MVs) was gathered: $V_l$, $AsNC_l$, $cArea_{l,I}$, and $Vs_{l,i}$. To identify linear relationships among the MVs, factor analysis (FA) ((rows) X (columns); (*l*) X (MVs)) was applied. The factor loadings per factor ($F_n$) indicated the weight of each variable to the corresponding factor while the factor scores per factor (*Fscores*) are the linear result of the initial variables with respect to this factor [49]. Due the fact that the MVs showed a non-normal distribution (Shapiro–Wilk W test < 0.90; $p < 0.05$), before the PCA application, the MVs were log-transformed (log-MVs) [46].

The *l* values that create the common $cArea_{3000}$ are members of the spatial cluster *j* ($sc_j$) of cage arrays while the maximum distance between *l* for membership is 6 km. For $sc_j$, the mean value (*mFscores*) and standard deviation (*SDFscores*) of the factor scores were estimated.

To examine the similarities of the *Fscores* between the *sc*, a clustering technique based on the Ward linkage and squared Euclidean distance ((rows) X (columns); (*sc*) X (*mFscores, SDFscores* of $F_n$)) was used.

Levene's test was applied to test for significant differences between the variance of the variables: *l*, Dst, $cArea_{3000}$, *mFscores*, and *SDFscores* among *sc groups*. In cases where the variance was not statistically significant (Levene's test; $p > 0.05$), analysis of variance (ANOVA; $p = 0.05$) to test for significant differences in these variables among *sc groups* and Bonferroni test were applied to check which *sc groups* differed from each other. In cases where the variance was statistically significant (Levene's test; $p < 0.05$), the Kruskal–Wallis test was used to check for significant differences between these variables among the *sc groups* and the Mann–Whitney test was sued to check which *sc groups* differed from each other [46].

As the legislation (Common Ministerial Decision No 31722/2011, FEK 2505 ratified on 4 November 2011) and Areas Organized for Aquaculture Development (AOAD) are defined, more or less, on the concept of the Allocated Zone for Aquaculture (AZA) [40,50], the existing fish farming site units fall under AOAD. The AOAD is classified according to

the priorities and the type of development into five categories: (A) particularly developed areas requiring improvement, modernization of farms and infrastructure, and better environmental management; (B) areas with significant scope for development; (C) remote areas with significant scope for development; (D) areas with particular environmental sensitivity requiring the adaptation of existing farms to the specific characteristics of the aquatic environment; and (E) suitable areas for further development of aquaculture (includes islands and shorelines with low farming activity).

Mapping on the coordination reference system ETRS89LAEA-ETRS89 Lambert Azimutal Equal Area was projected.

## 3. Results

In total, 433 fish farm cage arrays were recorded along the Greek coastline during the period 2016–2017 (Figure 1). The mean (±SD) farming volume of the arrays was $38.62 \pm 26.88 \times 10^3$ m$^3$, the mean number of squared cages was $5.43 \pm 10.22$ cages, the mean number of circular cages was $16.44 \pm 12.69$ cages, and the mean shorter distance of the cage arrays from shoreline was $154.55 \pm 177.23$ m. The total farming volume of the arrays was $16,726.6 \times 10^3$ m$^3$ while the total number of cages was 9977 (2355 squared and 7122 circular cages) (Table 2).

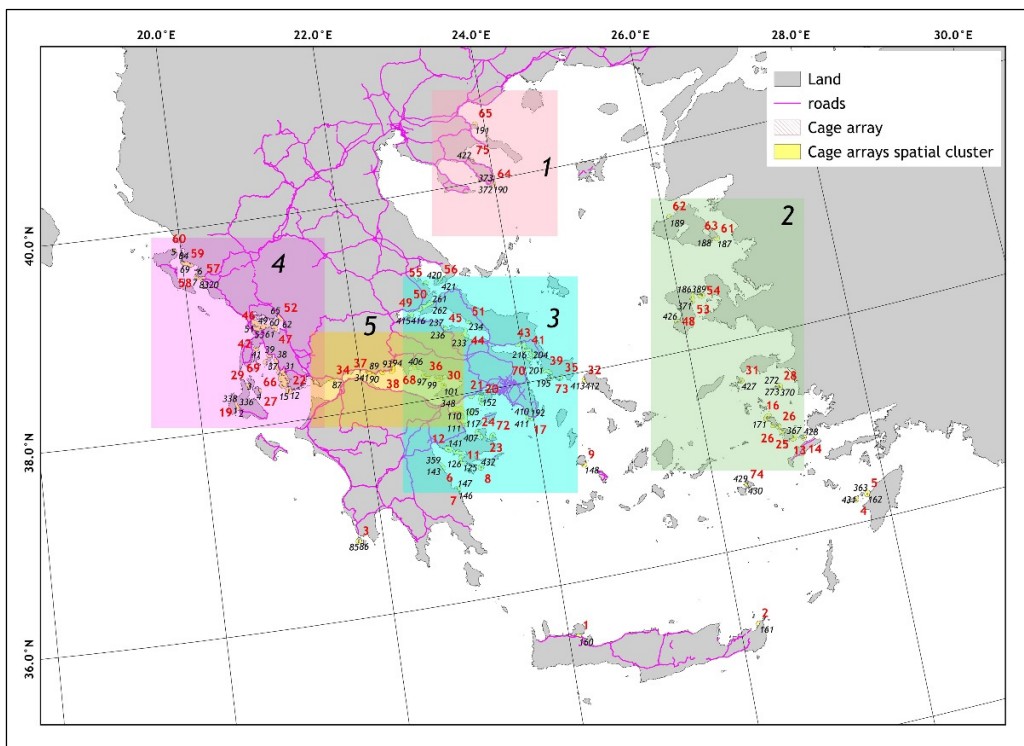

**Figure 1.** Mapping of the cage arrays' location and cage arrays' spatial cluster of Greek marine fish farms. Higher black numbers (1–5) mark the sites with an aggregation of cage arrays, red numbers show the cage arrays' spatial clusters (1–75), and lower black numbers the cage arrays (1–433).

In total, 1926 *cAreas* were recorded. The number of *cAreas* ranged from 75 to 433 (at $i = 3000$ and 0, respectively) (Figure 1). The mean *cArea* (±SD) ranged from $0.015 \pm 0.013$ to $32.78 \pm 29.52$ km$^2$, the mean number of cage arrays (*l*) ranged from $1.0 \pm 0.00$ to $5.77 \pm 8.65$ arrays and the total *cArea* per *i* ranged from 6.51 km$^2$ to 2469.15 km$^2$, at $i = 0$ and 3000 m, respectively (Table 3).

**Table 2.** Descriptive statistic of the cage arrays variables. Mean: mean value, SD: standard deviation, *NC*: number of circular cages, *NS*: number of square cages, *V*: functional farming volume of cages array, ANCS: arcsine-transformation of ratio of *NC* to total cages, Dst: shorter distance of cages array from shoreline, *l*: Number of cages array.

|  | **Mean** | **SD** | **Sum** | |
|---|---|---|---|---|
| $NC_l$ | 16.44 | 12.69 | 7122 | 9477 |
| $NS_l$ | 5.43 | 10.22 | 2355 | |
| $ANCS_l$ | 1.18 | 0.54 | - | - |
| $V_l$ ($\times 10^3$ m$^3$) | 38.62 | 26.88 | 16,726.6 | - |
| Dst (m) | 154.55 | 177.23 | - | - |
| $l$ | 433 | - | - | - |

**Table 3.** Mean value (standard deviation) and number (N) of *cArea*, numbers of arrays *(l),* and the functional farming volume per unit of impacted area (*Vs_l*) per impacted zone *i*.

| Impacted Zones *i* (m from Array's Edge) | N | $cArea_{i,j}$ (km$^2$) | $l$ | $Vs_l$ (m$^3$/m$^2$) | Sum of $cArea_{i,j}$ (km$^2$) |
|---|---|---|---|---|---|
| 0 | 433 | 0.015 (0.013) | 1.00 (0.00) | 3.80 (2.49) | 6.51 |
| 25 | 363 | 0.04 (0.04) | 1.19 (0.62) | 1.08 (0.51) | 16.36 |
| 50 | 308 | 0.10 (0.13) | 1.40 (0.96) | 0.56 (0.32) | 33.04 |
| 100 | 275 | 0.13 (0.06) | 1.57 (1.14) | 0.27 (0.17) | 37.26 |
| 200 | 230 | 0.31 (0.18) | 1.88 (1.47) | 0.11 (0.08) | 73.37 |
| 500 | 140 | 1.42 (1.17) | 3.09 (3.29) | 0.02 (0.02) | 199.87 |
| 1000 | 102 | 4.77 (3.91) | 4.24 (5.09) | 0.007 (0.008) | 487.44 |
| 3000 | 75 | 32.78 (29.52) | 5.77 (8.65) | 0.0009 (0.0010) | 2459.15 |
| total | 1926 | - | - | - | - |

Factor analysis extracted five factors (F1, F2, F3, F4, and F5) explaining 83.69% of the initial variance (Table 4). Using a cut-off value of ±0.50 for the factor loadings, F1 was positively associated with the $cArea_0$, $V_l$, and $Vs_i$ for $i > 0$, describing the *sc*'s intensity of impacts. The F2 was positively associated with $cArea_{500}$, $cArea_{1000}$, and $cArea_{3000}$ and negatively associated with $Vs_{500}$, $Vs_{1000}$, and $Vs_{3000}$, describing the high-distance (>200 m) impacted areas and the intensity of impacts. F3 was positively associated with $cArea_{50}$, $cArea_{100}$, and $cArea_{200}$, describing the mid-distance (50–200 m) impacted areas. F4 was positively associated with $cArea_0$ and ANCS and negatively associated with $Vs_0$, describing the local (under cages $i = 0$ m) impacted area and its intensity of impact and type of cages. Finally, F5 was positively associated with $cArea_{25}$ and negatively associated with $Vs$ ($Vs_{25}$), describing the low-distance ($i = 25$ m) impacted area and its intensity of impact.

The cluster analysis applied to *sc* was composed of more than one cage array while this group made the *sc group 1*. The cluster analysis of the *Fscores* revealed another five groups of cage array spatial clusters (*sc group 2 . . . sc group 6*). The mean number of arrays differed among the *sc groups* (Kruskal–Wallis; $p < 0.05$). In *sc group 1*, there were 27 spatial clusters; in *sc group 6*, there were 15 spatial clusters; in *sc groups*, there were 2 and 3 and 11 and 10 spatial clusters; and in *sc groups 4* and *5*, there were 7 and 5 spatial clusters, respectively (Table 5).

**Table 4.** Factor loadings, explained variance (%ExpVar), and cumulative explained variance (%CExpVar) of the extracted factor (F1–F5) for the log-transformed variables (log- MV; $cArea_i$: common area of *ls* for distance *i*, $Vs_i$: impact intensity for distance *i*, $V_l$: functional farming volume, ANCS: arcsine-transformation of ratio of NC to total cages). Bold marks the important values (cut-off value of $\pm 0.50$).

| | Factors | | | | |
|---|---|---|---|---|---|
| **Log−MV** | **F1** | **F2** | **F3** | **F4** | **F5** |
| $cArea_0$ | **0.601** | 0.037 | 0.183 | **0.705** | 0.183 |
| $cArea_{25}$ | 0.378 | 0.088 | 0.291 | 0.353 | **0.712** |
| $cArea_{50}$ | 0.086 | 0.022 | **0.737** | 0.073 | 0.483 |
| $cArea_{100}$ | −0.043 | 0.065 | **0.908** | 0.080 | 0.171 |
| $cArea_{200}$ | −0.116 | 0.303 | **0.785** | 0.229 | −0.209 |
| $cArea_{500}$ | −0.075 | **0.756** | 0.320 | 0.225 | −0.115 |
| $cArea_{1000}$ | −0.006 | **0.876** | 0.136 | 0.177 | 0.038 |
| $cArea_{3000}$ | 0.081 | **0.782** | 0.076 | −0.027 | 0.180 |
| $Vs_0$ | −0.094 | −0.018 | −0.052 | **−0.902** | −0.168 |
| $Vs_{25}$ | **0.710** | −0.034 | 0.015 | −0.084 | **−0.595** |
| $Vs_{50}$ | **0.868** | −0.001 | −0.235 | 0.187 | −0.270 |
| $Vs_{100}$ | **0.913** | −0.033 | −0.277 | 0.167 | −0.004 |
| $Vs_{200}$ | **0.876** | −0.183 | −0.245 | 0.065 | 0.217 |
| $Vs_{500}$ | **0.700** | **−0.563** | −0.030 | 0.030 | 0.174 |
| $Vs_{1000}$ | **0.629** | **−0.672** | 0.113 | 0.054 | 0.093 |
| $Vs_{3000}$ | **0.566** | **−0.649** | 0.165 | 0.171 | 0.027 |
| $V_l$ | **0.892** | −0.016 | 0.273 | 0.207 | 0.163 |
| ANCS | 0.151 | 0.124 | 0.130 | **0.736** | −0.025 |
| %ExpVar | 30.17 | 18.32 | 14.19 | 12.63 | 8.38 |
| %CExpVar | 30.17 | 48.49 | 62.68 | 75.31 | 83.69 |

**Table 5.** Spatial cluster's functional volume ($V_l$), mean number of cage arrays, mean shorter distance of cage arrays from the shoreline (Dst), mean $cArea_{3000}$ (SD: standard deviation), number of spatial clusters of cage arrays (*sc*), and total $cArea_{3000}$ per *sc* groups. Superscript letter: homogeneity groups from ANOVA on log-transformed data; Bonferroni test: $p < 0.05$; Superscript number: homogeneity groups from Kruskal–Wallis and Mann–Whitney test: $p < 0.05$.

| | Sc Groups | | | | | | Total |
|---|---|---|---|---|---|---|---|
| | 1 | 2 | 3 | 4 | 5 | 6 | |
| $V_l$ [×10³ m³] (%) | 1338.13 (8.00%) | 2083.01 (12.45%) | 1546.3 (9.24%) | 602.31 (3.6%) | 574.25 (3.43%) | 10,582.55 (63.27%) | 16,726.56 |
| mean number of cage arrays (range) | 1 (1–1) [1] | 3.63 (2–7) [2] | 3.9 (2–8) [2] | 5.14 (2–11) [2] | 2.2 (2–3) [2] | 18.6 (5–48) [3] | 5.77 (1–48) |
| *l* | 27 | 40 | 39 | 36 | 11 | 280 | 433 |
| $cArea_{3000}$ (SD) [km²] | 17.86 (4.72) [a] | 31.18 (8.37) [b] | 36.37 (10.40) [b] | 22.84 (11.20) [a] | 17.05 (2.56) [a] | 68.31 (49.50) [c] | 32.78 (29.50) |
| Dst [m] | 110.11 (72.71) [1,2] | 140.32 (129.41) [1,2] | 90.59 (42.06) [1] | 96.65 (38.38) [1] | 119.71 (43.02) [1,2] | 178.59 (208.46) [2] | 154.55 (177.23) |
| Number of *sc* | 27 | 11 | 10 | 7 | 5 | 15 | 75 |
| total $cArea_{3000}$ [km²] | 482.48 (19.62%) | 343.04 (13.95%) | 363.75 (14.79%) | 159.89 (6.5%) | 85.26 (3.47%) | 1024.73 (41.67%) | 2459.15 |

The *sc group 6* has a greater number of cage arrays (5–48), followed by the *sc groups 2, 3, 4,* and *5* (2–11 cages arrays) while the *sc group 1 has* the lowest number of cage arrays (1 cages array) (Mann–Whitney test: $p < 0.05$). The mean $cArea_{3000}$ of *sc group 6* was greater at 68.31 ($\pm$49.50) km² than *sc group 2* and *sc group 3* (31.18 $\pm$ 8.37 and 36.37 $\pm$ 10.40 km², respectively) and also greater than *sc group 1, sc group 4,* and *sc group 5* (17.86 $\pm$ 4.72, 22.84 $\pm$ 11.20, and 17.05 $\pm$ 2.56 km², respectively) (ANOVA on log-transformed data; $p < 0.05$; Bonferroni test; $p < 0.05$). The mean shorter distance of the cage arrays from the shoreline was higher in the *sc group 6* (178.59 $\pm$ 208.46 m). Most of the functional farming volume was covered by the *sc group 6* (63.37%) while the other *sc groups* shared 3.43% (*sc group 5*) to 12.45% (*sc group 2*) of the total functional farming volume (Table 5).

The ANOVA showed statistically significant differences regarding the *mFscores* between the *sc* groups ($p < 0.05$). The Bonferroni test indicated for F1 *Fscores*:

[*sc group 4*] < [*sc group 6 = sc group 3 = sc group 5*] < [*sc group 5 = sc group 1 = sc group 2*]

for F2 *mFscores*:

[*sc group 1 = sc group 5*] < [*sc group 4 = sc group 2*] < [*sc group 2 = sc group 3*] < [*sc group 6*]

for F3 *mFscores*:

[*sc group 1 = sc group 2 = sc group 3 = sc group 4*] < [*sc group 5 = sc group 6*]

for F4 *mFscores*:

[*sc group 4*] < [*sc group 2 = group 6 = sc group 1 = group 5*] < [*sc group 1 = group 5 = sc group 3*]

for F5 *mFscores*:

[*sc group 2*] < *sc group 1 = sc group 2 = sc group 3 = sc group 4 = sc group 5 = sc group 6*] < [*sc group 6*] (Bonferroni test; $p < 0.05$) (Table 6).

**Table 6.** One-way ANOVA results of the mean *Fscores* (*mFscores*) and Kruskal–Wallis of the standard deviation (*SDFscores*) per factor (1 to 5) results, according to the spatial cluster group (*sc groups*). */ns: statistically/non statistically significant difference ($p < 0.05$), respectively; np: *sc group* non participated in ANOVA. Horizontal, the same number marks ascendingly according to mean value, the homogeneous groups (Bonferroni test; $p < 0.05$ for *mFscores* and Mann–Whitney test; $p < 0.05$ for *SDFscores*).

| | Description | Sc Groups | | | | | | |
|---|---|---|---|---|---|---|---|---|
| | | **1** | **2** | **3** | **4** | **5** | **6** | *ANOVA* |
| F1 *mFscores* | *sc's intensity of impacts* | 3 | 3 | 2 | 1 | 2, 3 | 2 | * |
| F1 *SDFscores* | *Heterogeneity of sc's intensity of impacts* | np | 1 | 1 | 1 | 1 | 1 | ns |
| F2 *mFscores* | *High distance (i > 200 m) impacted areas and their intensity of impacts* | 1 | 2, 3 | 3 | 2 | 1 | 4 | * |
| F2 *SDFscores* | *Heterogeneity of high distance (i > 200 m) impacted areas and their intensity of impacts* | np | 1 | 1 | 1 | 1 | 1 | ns |
| F3 *mFscores* | *Mid distances (50 < i ≤ 200 m) ipmacted areas* | 1 | 1 | 1 | 1 | 2 | 2 | * |
| F3 *SDFscores* | *Heterogeneity of mid distances (50 < i ≤ 200 m) impacted areas* | np | 1, 2 | 1, 2 | 1 | 1 | 2 | * |
| F4 *mFscores* | *Local impacted area (under cages: i = 0 m), its intensity of impactand type of cages* | 2, 3 | 2 | 3 | 1 | 2, 3 | 2 | * |
| F4 *SDFscores* | *Heterogeneity of local impacted area (under cages: i = 0 m), its intensity of impact and type of cages* | np | 2, 3 | 1 | 3 | 1, 2 | 1, 2 | * |
| F5 *mFscores* | *Low distance (i = 25 m) impacted area and its intensity of impact* | 1, 2 | 1 | 1, 2 | 1, 2 | 1, 2 | 2 | * |
| F5 *SDFscores* | *Heterogeneity of low distance (i = 25 m) ipmacted area and its intensity of impact* | np | 1 | 1 | 1 | 1 | 1 | ns |

The Kruskal–Wallis test showed statistically significant differences regarding the *SDFscores* between the sc *groups* ($p < 0.05$) only for F3 *SDFscores* and F4 *SDFscores*. The Mann–Whitney test indicated for F3 *SDFscores* that:

[*sc group 4 = sc group 5*] < [*sc group 2 = sc group 3 = sc group 4 = sc group 5*] < [ *sc group 6*]

and for F4 *SDFscores*:

[*sc group 3*] < [*sc group 3 = sc group 5*] < [ *sc group 2 = sc group 5 = sc group 6*] < [*sc group 4*]

It is noted that in the Kruskal–Wallis test, the *sc group 1* was excluded as the *SDFscores* for all *Fs* were 0 (Table 6).

The 5 more extended *scs* (belonging to *sc group 6*) covered 35.5% of the total *V* and located in west-central Greece (ID: 66; 204 km$^2$; 11.2% of total *V*), Saronikos Gulf (ID:15;142 km$^2$; 7.9% of total *V*), Amvrakikos Gulf (ID: 46; 109 km$^2$; 5.9% of total *V*), South Evoikos Gulf (ID: 41; 85 km$^2$; 5.9% of total *V*), and North Evoikos Gulf (ID: 50; 73 km$^2$; 5.9% of total *V*) (Figure 2).

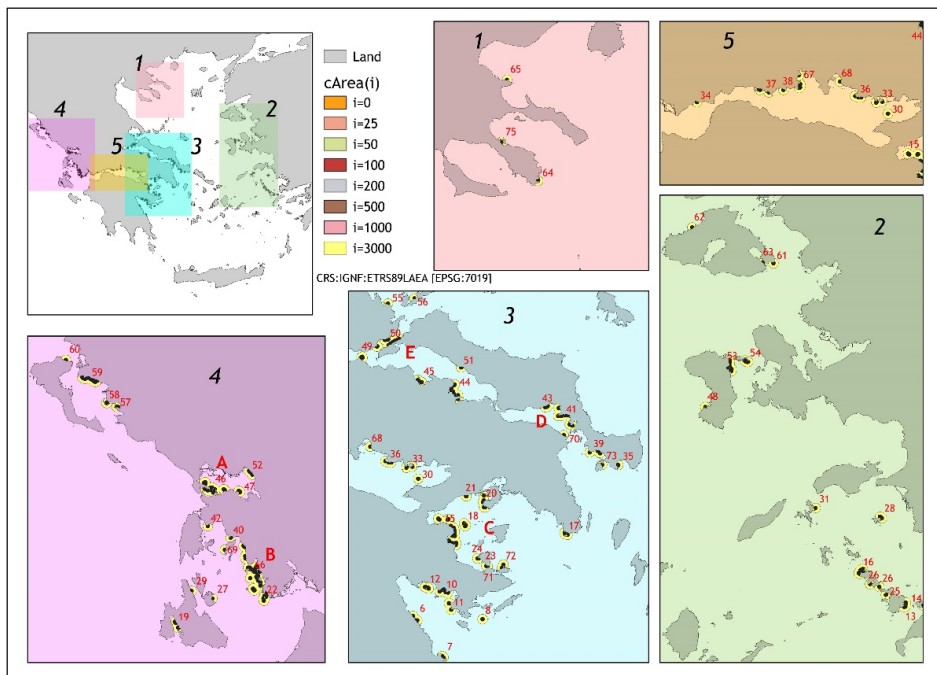

**Figure 2.** Mapping of the impacted area (*cArea*) for various distances (*i* = 0: under cage array, *i* = 25: area up to 25 m from cages' edge . . . *i* = 3000 area up to 3000 m from cages' edge). A: Amvrakikos Gulf, B: West-Central Greece, C: Saronikos Gulf, D: South Evoikos Gulf, E: North Evoikos Gulf. The numbering of the maps is similar to that in Figure 1.

The greatest farming volume was recorded for the AOAD category A (71.86% of the total volume) with the dominant contribution of *sc group 6* (54.11% of total volume) while the farming volume contribution of the other AOAD categories ranged from 0.57 to 13.21% of the total volume (Figure 3).

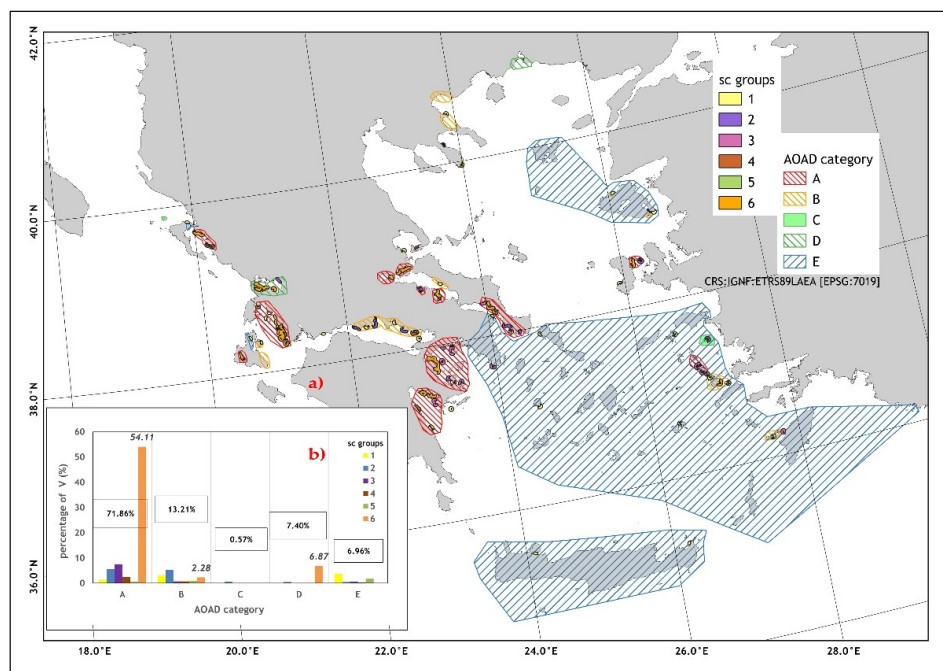

**Figure 3.** Mapping of the spatial cluster groups (*sc groups*) of Greek marine fish farms and Areas Organized for Aquaculture Development (AOAD) categories (**a**) and the distribution of the farming volume (V%) according to the AOAD categories and *sc groups* (**b**).

## 4. Discussion

The seabass and seabream culture in Greece began in the early 1980s, and after 1995, it entered a stage of maturation. This stage, at a technological level, was best known for the trend of increased production volume by changing the size and shape of the cages being used (from squared and cyclical cages with s perimeter < 40 m to cyclical cages with a perimeter > 40 m) [40,51]. A study based on satellite images for the period 2001–2011, recorded a considerable change toward larger cages accompanied by the uninstalling and/or relocation by new installations of farms [44]. In the present study, for the period 2016–2017, along the Greek shoreline, 9477 cages were recorded. This result is in line with a recent study [37] for June 2016 while a previous study [31] in 2006 recorded 10,422 cages. This difference is somewhat expected due to the inter-temporal modification of farm structures from small to larger cages [41,51] to increase the production volume. According to our findings, the independence of the cyclical cages' number percentage of cage arrays (ANCS) by the functional farming volume ($V_l$) (Table 4: F1) indicates that the modification of farm structures is reaching the climax phase.

The farming activity is located near the shoreline, with most of it (81.9% of the total functional farming volume) being located along the mainland shoreline and in particular in the central part of country (Figure 1), in close vicinity to the main road network. Farmed fish is an export-oriented product for major EU markets [42], so the small distance from the national highways to the major ports Igoumenitsa and Patras is an asset to the supply chain.

The area that may have been environmentally impacted by fish farming activity ranged from 6.51 km$^2$ (under the cages arrays: impact on bentho-communities [6–12]) to 2459.15 km$^2$ (zone 3000 m around the arrays: impact on nekton-communities demography) [13–18,52,53] (Table 1). According to the higher common zone of cage arrays, 75 spatial clusters were created (Figure 2). Inside the spatial cluster, the cage array members can be considered as a field of wild fish's attraction and concentration [13,14,16,25,28] since connectivity through wild fish is apparent [17] through the escape of farmed fish to the wild and vice versa [52]. Moreover, wild fish do not seem to participate in the transmission of diseases and parasites in the spatial cluster [54] for escaped fish, which is a true fact [53]. Thus, the spatial clusters should be considered as a delimited area for the research and management of diseases (namely, disease management area [55]) and of escaped fishes.

Greek legislation (Common Ministerial Decision No 31722/2011, FEK 2505 ratified on 4 November 2011) provides detailed definitions regarding the fish farm unit (owning entity), the park, and the arrays of cages. The distance among the parks of a unit can be between 100 and 250 m and higher than 500 m among the units. Moreover, the allowed carrying capacity (related to the functional volume) of a specific location (in practice, the leased area) is controlled by several parameters, such as the currents' velocity, geomorphology due to the openness/exposure of a location to open sea, bathymetry, and distance of thee farm from the shoreline, etc. (121570/1866/12 June 2009 common newsletter of Ministry Environmental, spatial planning and Ministry Rural development and foods of Greece) [7].

Although the legal regime is recent, Greek marine finfish farms operate under the legal specifications [7]. The above legal guidelines have led to a spatial pattern of cage arrays in which the short-distance zones ($i < 50$) are impacted, absolutely, by one cage array while the mid-distance (50–500 m; $i$: 50–500) and long-distance zones (>500 m; $i > 500$) are impacted by cage arrays belonging to different parks of the same unit and by different units, respectively. This latter finding is based on a study of 230 farm units (equal to the number of $cArea_{200'}s$) comprising $1.88 \pm 1.47$ cage arrays (Table 3), which is an estimation is close to the number of Greek finfish marine farms that was recorded in 2013 (240 farms) [7].

The factor analysis revealed that the mean functional volume of the cage arrays of a spatial cluster controlled the intensity of impacts at the studied distances (at least for distances $\geq 25$ m; $i \geq 25$) (Table 4: F1). F2, F4, and F5 support an inversely proportional relationship between $cAreas_{500–3000}$, $cAreas_0$, and $cAreas_{25}$ and their impact's intensity, respectively, while $cAreas_{50–200}$ was not related to its impact's intensity (F3). For the cases of $i = 0$ and $i = 25$, the inverse relationship between $cAreas$ and $Vs$ is expected as the

cage's establishment in a cage array occurs at a given distance among them, and so, by increasing $V_l$ of the array via the addition of cages, this increases *cArea*. In the case of $i = 0$, it seems that the usage of cyclical cages, for a given $V_l$, increases *cArea$_0$* due to their mooring technique, which requires a greater distance among them than for squared cages. Regarding *cArea$_{500-3000}$*, it is easy to estimate the integration of a cage array belonging to another farm unit (minimum distance between units $\geq$ 500 m), which increases *cArea$_{500-3000}$* exponentially, resulting in an inverse relationship between *Vs* and *cArea*.

On the other hand, keeping in mind the shorter distance from the shoreline for most of the cage arrays is 50–200 m (Table 2), the variation in the local shoreline geomorphology affects the size of *cAreas$_{50-200}$* and a positive relation between the local shoreline geomorphology variation and the variance of *cAreas$_{50-200}$* is expected, driving their independency by their own *Vs*. Furthermore, the relatively low accuracy of the shoreline capturing the land layer (around to 10% at $i = 50$ to 3% at $i = 200$; estimated as the error zone that is 2 times the accuracy of the land layer by a width of 50 m multiplied by the 100 m length of shoreline; Table 3) in relation to the size of *cAreas* increase the variance of *cAreas*. Finally, these zones refer to the cage arrays that comprise a unit. As per the abovementioned factors, the carrying capacity (related to the functional volume) of a specific location is controlled by several parameters [7], which lead to an expected variation of $V_l$ for given *cAreas$_{50-200}$*.

The aquaculture industry operates on the condition that a particular regulatory framework is used, with provision of an Allowable Zone of Effects (AZE) as has been used by Scottish Environment Protection Agency (SEPA) in Scotland [56], where specific environmental quality standards pertain. This is true considering that specific environmental quality standards in the zone up to 100 m from the cages' edges in the direction of the common currents are allowed (121634/7242/20 December 2019 newsletter of Ministry Environmental and Energy). Thus, *cArea$_{100}$*, as estimated in the present study, could be considered as the AZE according to the allowed specific environmental quality standards as defined by Greek legislation. Additionally, numerous fish farms participate in ASC's (Aquaculture Stewardship Council's) aquaculture certification program for recognition and to gain the reward of responsible aquaculture. From its standards comes the definition of AZE, which is defined as a zone of 25 m when the AZE has not been defined using a robust and credible modeling system [43].

The 15 *sc* values (out of 75 *sc*) that constitute the *sc group 6* cover 63.2% of the total functional farming volume (Table 5), indicating that Greek marine fish farming activity shows a high level of spatial aggregation. On the other hand, the majority of fish farming activity showed a relative moderate intensity of impacts (*sc groups 3, 5*, and *6*; cover 75.9% of the total functional farming volume (Table 5). After six years from the legislation application, the distribution of farming activity among the AOAD categories seemed to not have changed (as well as the production [41]), which may be due the deceleration of the farming activity growth rate, thanks to the Greek debt crisis [40].

Locations with a high clustering of farms showed a notable increase in small-scale fishery production [28]. The fact that these zones belong to the coastal zone of regions that show a relative dependence on small-scale fishery [57] indicates a possible important conflict for sites in the vicinity of farms and synergy for sites out of farms' vicinity, among the two activities (Figure 4). On the other hand, the presence of extensive fish farming activity (*sc* ID: 66, 22) in the vicinity of Mesolongi-Aitoliko lagoons (protected habitat: Natura 2000) and in Amvrakikos Gulf and in the vicinity of their lagoons (also protected area: Natura 2000) (*sc* ID: 46, 47, and 52) has played a crucial role not only in lagoonal ichthyofauna biodiversity changes [58] but also in the reduction of fishery production of the lagoon [59,60], affecting the functions of the lagoon habitats (Figure 4).

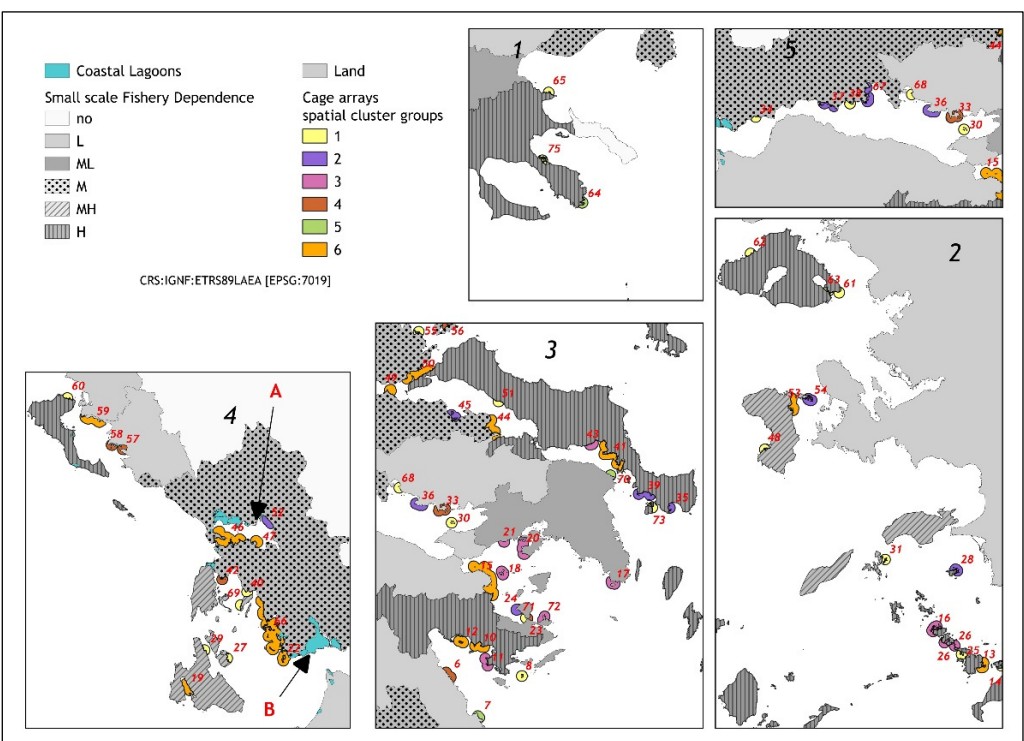

**Figure 4.** Mapping of spatial cluster groups of Greek marine fish farms, small fishery dependence classification, and coastal lagoons. A: Amvrakikos Gulf, B: Mesolongi-Aitoliko lagoons, no, L, ML, M, MH, H: non, Low, Moderate Low, Moderate, High moderate and High Small-scale fishery dependance, respectively. The numbering of the maps is similar to that in Figure 1.

A number of factors controlling the accuracy of the shape and size of the zones and the intensity of impacts are: (a) the hydrodynamic conditions of each location play a key role in the dispersion of waste [6,26,61–63]; (b) in the present study, the impact zones were estimated as a projection on the sea surface so the actual impacted area (especially that on the substrate: $i = 0$ up $i = 500$ m) of the bottom is expected to be greater than was estimated due to the deformations of the bottom terrain, with small variations in the bathymetry resulting in significant changes in the sedimentation pattern [64]; (c) the assumption that the intensity of impacts only depends on the functional volume while various patterns of feeding occur among farms [65,66]; (d) changes in the cage array locations (relocation, add or removal) can change the impact zones sizes; (e) the accuracy of the land layer affects the size estimation accuracy of the mid-distance impact zones. An improvement would be to estimate the shoreline using satellite images with a higher accuracy (i.e., Sentinel 2; spatial resolution of 10 m [67]) or to perform manual correction following the coastline on the Google Earth images; and (f) the progressive decrease in the impact intensity according to the distance from the cage arrays (see the citations in Table 1) was not considered. An improvement is to decrease the grid cell size and the cell's weighting according to the distance.

## 5. Conclusions

The present study aimed to provide a detailed record of marine fish farms on the Greek coastline, the spatial distribution of the possible environmental impacts, and a mapping and evaluation of their spatial variability. The study aids in the enhancement of georeferenced information about Greek aquaculture. The findings revealed that Greek marine fish farming activity shows a high level of spatial aggregation but with a relative moderate intensity of impacts as a result of the legal constraints, which play a crucial role in controlling the spatial distribution of activity at a local, regional, and national scale. Although the data

refer to the period 2016–2017 and an update is necessary, the excepted changes nonetheless do not refer to most of the installed activity. Although further development activity is expected in the AOAD categories B and E, the produced vectors could be used, as the case may be a baseline to solve local and regional problems after an update of the interested region.

The next update of the GIS database should also incorporate the leasing area, cages with the other facilities including the land of units, their monitoring programs, and other information (climatic, physicochemical, currents, bathometry, etc.). This will be a very useful tool, which is expected to have a notable contribution to the mid-term challenges of the sector as the Growth Vision 2030 for Greek marine fish aquaculture is in progress [42]. This challenge will require more space and it will be found to oppose land use competition, legal restrictions about the environment (Common Ministerial Decision No 31722/2011, FEK 2505 ratified on 4 November 2011), and official funding motivations for growing aquaculture that provide environmental services (EU Regulation No 508/2014; art.54).

**Author Contributions:** Conceptualization, G.K. and K.T.; methodology, G.K.; software, G.K.; validation, G.K., K.T. and J.A.T.; formal analysis, K.T.; investigation, K.T.; resources, G.K. and J.A.T.; data curation, G.K.; writing—original draft preparation, K.T.; writing—review and editing, G.K. and J.A.T.; visualization, G.K. and K.T.; supervision, G.K.; project administration, J.A.T.; funding acquisition, J.A.T. All authors have read and agreed to the published version of the manuscript.

**Funding:** This research was funded by the "Innovation in Aquaculture" EU-Greece Operational Program of Fisheries, EPAL 2014–2020 grant number (MIS) 5067321, as a part of the project "Improving competitiveness of the Greek fish farming through development of intelligent systems for disease diagnosis and treatment proposal and relevant risk management supporting actions".

**Institutional Review Board Statement:** Not applicable.

**Informed Consent Statement:** Not applicable.

**Data Availability Statement:** Data sharing not applicable.

**Acknowledgments:** Special thanks to Cpt. Grigorios Tsolakos, for assisting in drafting this manuscript.

**Conflicts of Interest:** The authors declare no conflict of interest.

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
