# Peer review of "Mapping of Greek Marine Finfish Farms and Their Potential Impact on the Marine Environment"

_jmse, doi:10.3390/jmse10020286_

Round 1

Reviewer 1 Report

This paper provides a detailed record of marine fish farms in 341 Greek coastlines, the spatial distribution of the possible environmental impacts, mapping and 342 evaluation of their spatial variability. This is an interesting study but the following concerns should be addressed before I recommend it for publication.

major concerns:
1) conclusions of your study
I did not find any obvious conclusion from your results and discussions, what are the findings of your study apart from the factual numbers you reported?

2) time windows of the remote sensing images
The authors state they used the Google Earth satellite images for the period of 54 June 2016 to May 2017 with fish farms cage arrays, however, the season variation is a factor that may cause some error detections. When you design the experiment, did you make sure to map the cages in relative consistency time windows, i.e. summer images from 2016 and summer images in 2017?

3) Introduction
The introduction is very weak and did not address any recent studies on this topic, you should extend this part to a reasonable length and fully discuss the problems facing and your motivation to solve them.

4) English writing
General communications are poor,  for example, just the last sentence in the abstract just got a 2 grammar mistakes. The results reflect to (to is redundant)an ......the monitoring of environment (environmental)....

minor concerns:
1) page2 line 63-80, QGIS is just a tool, you should address its key functions more instead.
2) page4 line110, the mean circular cages 16.44?
3) page5 line 129,  five factors (F1, F2, F3, F4 & F5),  better describe this more specifically.
4) page13 line 337 as (an) adaptation?
There are other grammar mistakes, please read them carefully when resubmitting.

Reviewer 2 Report

Abstract 

Please define GPD 

Introduction 

References should be updated 

L32 instead of GIS, or in parallel with GIS, I will add something about the increasing use of remote sensing images for aquaculture application (see for example the special issue of Frontiers in Marine Science “Remote Sensing for Aquaculture” (https://www.frontiersin.org/research-topics/9293/remote-sensing-for-aquaculture#articles) 

The introduction should be extended and the context explained. 

Materials and Methods 

I think that the description of QGIS command is not necessary, these are very common operations. Removing this part can help to have a more readable text. Moreover, too many abbreviations are reported, and it is hard to follow the methodology and result sections. 

L85-86 Why did you log-transformed the data? 

Results 

L96-L104 this paragraph is hard to follow, please rephrase. 

L109 & L112– 26.88x? 

L142-143 Add Table 5. 

Tables 2, 3 and 4. Please define the abbreviations in the caption. 

Table 5. Please add an extra space between the row “V1...(%)” and the row “mean number of cages.....(ranges)”. 

L182 - Table 6 caption. Correct value. 

Discussion 

L207-209 This result is also in line with the same analysis carried out in Sarà et al., 2018 (see Figure S2).  

Sarà, G., Gouhier, T. C., Brigolin, D., Porporato, E. M., Mangano, M. C., Mirto, S., ... & Pastres, R. (2018). Predicting shifting sustainability trade‐offs in marine finfish aquaculture under climate change. Global change biology, 24(8), 3654-3665. 

L220 – 227. In my opinion, this section, including Fig 3 are results. Moreover, the small fishery dependence classification areas are reported here for the first time and no explanation about how this area are selected. The letter reported in the legend are not detailed in the figure caption.  

L293-314 This section should be moved to material and methods and results paragraphs. 

L319-324 Please add the locations reported in the text in the map, in Fig.2 are not reported the names of the Gulfs nor the protected areas.  

Figure 4. The graph inside the map is too small. 

Round 2

Reviewer 1 Report

line 159-160, it is a bit strange, maybe try to merge it into the above paragraph.
All figures seems to be produced in low resolution, please update with a full resolution image.